# Customer Experience in Circular Economy: Experiential Dimensions among Consumers of Reused and Recycled Clothes

An Hai Ta, Leena Aarikka-Stenroos * 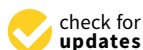 and Lauri Litovuo

Unit of Industrial Engineering and Management, Faculty of Management and Business, Tampere University, Korkeakoulunkatu 7, 33720 Tampere, Finland; hai.ta@relexsolutions.fi (A.H.T.); lauri.litovuo@tuni.fi (L.L.)
* Correspondence: leena.aarikka-stenroos@tuni.fi; Tel.: +358-503-015-476

**Abstract:** The textile and clothing industry is undergoing a sustainability transition, pushing related businesses to adapt to circular economy (CE) models, such as recycling and reuse. This shift has been extensively studied from industry and business model perspectives, but we lack an understanding of the customer perspective, i.e., how circulated products, such as reused and recycled clothes are experienced among consumers. This understanding is crucial, as customer experience plays a significant role in the adoption of CE products. Therefore, we conducted a qualitative interview study to explore how consumer-customers experience recycled textiles and reused clothes. We used an established experience dimension model and mapped how the five dimensions of customer experience—sensory, affective, behavioral, cognitive, and social—present themselves in the sustainable clothing industry. The data comprised 16 qualitative semi-structured interviews analyzed with a coding framework built on the basis of customer experiences, customer values, and the CE business model literature. The results revealed that diverse sensory (e.g., scent), affective (e.g., pride and shame), behavioral (e.g., developing new decision-making rules), cognitive (e.g., learning and unlearning), and social (e.g., getting feedback from others and manifesting own values) aspects shape how consumers experience reused and recycled clothes. We also compared and analyzed the results of the reuse and redistribute model and the recycle model. Our study contributes to the literature of CE business models and customer experience by providing a structured map of diverse experiential triggers and outcomes from the five experiential dimensions, which together reveal how consumers experience circulated products of the clothing industry. These findings enhance our understanding of customers' motivation to use recycled and reused products and adoption of CE products.

**Keywords:** circular economy; clothing; textile and fashion industry; consumer experience; customer experience; recycle business model; reuse and redistribute business model

## 1. Introduction

The linear model of take, make, and dispose resources and materials has struck a devastating blow to the environment [1]. Consequently, the circular economy (CE) is rising in both policies and business [2,3] globally in different industry sectors, and in turn, product and service offerings from all types of CE business models, such as reuse, recycle, or reduce, are increasing and developing. This CE shift has led to individual companies increasingly providing more circular value propositions and pursuing competitive advantage from circularity in their products, services, and solutions [4], as well as industries jointly developing by increasing circulation in their supply–value chains, for example, the textile industry [5–8]. Consequently, consumers have started adopting and seeking more circular offerings [9]. Despite the evidenced shift toward CE in production and consumption, the focus has been on policies [10] and companies' business models [4], while neglecting customer-centric perspectives that focus on examining how customers consume and perceive circular, recycled, or reused offerings. The change from linearity to circularity, however, also alters the nature and many characteristics of products and

services, and consequently, how they are experienced by customers. Thereby, this paper brings a new theoretical lens, customer experience, into the examination of CE businesses. Customer experience, conceptualized as customers' subjective, contextual perception of an offering [11–13], has been identified as a compelling antecedent of competitive advantage in various business contexts [11]. Leading traditional brands, such as Dell, Starbucks, and Apple, have already posited customer experience at the center of their strategic and managerial focus and acknowledged the successful results of their efforts. Likewise, scholars have emphasized customer experience in achieving satisfied and loyal customers [14] who are willing to pay [15], thus making it pivotal for CE businesses as well. However, while the business and organizational aspects of CE have been focused on [4,16,17] and the importance of customers and their acceptance in establishing a successful circular business has been recognized [18,19], customer experience in CE has been neglected. While resolving resource issues and combatting the dominant linear business models, the companies in circular business must offer compelling customer value [20]; however, no focused studies have been conducted on customers' experiential perceptions of CE offerings. The studies examining customer adoption and acceptance [21,22] or consumer roles [23] of recycled or reused products do not provide this understanding. To fill this research gap, the research goal of this paper is to explore and map how customers experience CE products.

Customer experience has been increasingly studied in the traditional linear economy, in relation to different research streams such as service marketing [24], consumer research [25], online marketing [26], branding [27], and experiential marketing [28]. These studies have evolved along with the emergence of the so-called experience economy [29], meaning that products and services can excel by creating personal experiences that individual customers value. Based on the widely accepted dimension model of customer experience, the experience comprises five dimensions: sensory, affective, cognitive, behavioral, and social [27,28,30]. This dimension model enables researchers, marketers, and companies to understand how positive customer experiences are comprised and how they can be triggered and managed. The starting point of this study is the insight that these experiential dimensions are still highly relevant in CE, and a better understanding of all dimensions of customer experiences concerning CE products can help make these products more attractive to consumers. The benefits of delivering a superior customer experience are extensive: increased purchase intention [13], better customer loyalty and satisfaction [31], and positive word-of-mouth [29]. In particular, the provision of a superior customer experience is difficult to imitate by others and hence, provides a fruitful opportunity for circular businesses to gain a sustainable competitive advantage. Thus, creating a superior customer experience can increase awareness regarding CE businesses and sustainable consumption. As recycled and reused products inherently originate from different circular business models [32], we believe that customer experiences may differ consequently and will examine this aspect.

As the clothing, textile, and fashion industries are one of the most polluting industries globally [7] and consumers' behavior and adoption of more circulated products is crucial [9], we select the clothing industry as our research context to study the customer experience of CE offerings, particularly recycled and reused products. Research discussing the CE of the textile, clothing, and fashion industries has provided only an overview of the circularity-increasing changes in industry practices, incentives, business models, and value chains [5–8,33] and only rarely discussed customer roles [21,34].

To summarize the extant gaps in the relevant literature, the CE and environmental sustainability research has neglected customer perspective and particularly customer experience that would explain how circularity and circular products are experienced by the customers themselves, whereas customer experience research has implicitly focused to analyzing linear products without particular focus on circular products, and is therefore lacking specific understanding of how customer experiences are constructed and manifested in circular economy products.

Due to the abovementioned research gaps and the need for new knowledge, we pose the following research question: "How are CE products (recycled vs. reused) experienced by consumers in the clothing industry via the experiential dimensions?" To answer this question, we conduct a qualitative explorative interview study among consumers, map different dimensions of their experiences with recycled and reused clothes to examine how each five dimensions manifest, and then through comparison to synthesize a more general model that displays how circular products are experienced by consumers, for each of the five dimensions.

This paper makes the following contribution to the literature. First, it aims to primarily contribute to sustainability and CE research, by building a micro-consumer-level understanding of CE and by introducing the customer experience approach and the consequent subjective individuals' perspectives on CE business: it develops a conceptual map on how consumers experience circular offerings according to the five experiential dimensions. The contribution here is the model that displays how customers experience circular products through sensing, feeling emotions, cognitive processing, behaving, and interacting socially. Second, it develops detailed empirical knowledge of customer experience in two circular business models—recycle & reuse and redistribute—to explain how recycled and reused products differ from a customer's perspective.

## 2. Theoretical Background

### 2.1. Customer Experience and Its Dimensions

The understanding and theoretical development of the concept of customer experience has rapidly evolved recently in both research and practice. Customer experience, however, is a complex concept to be studied, as it can be approached from various angles [11,12]. Some researchers have approached customer experience as a perception that organizations can manage, design, or deliver, whereas others have argued that it is an individual's subjective and contextual perception [11]. There are also differing views on how customer experience emerges and who can contribute to it: some studies have argued that customer experience is created dyadically between a firm and a customer [30], whereas others have argued that it is developed through various interactions between diverse involved actors [11]. Some studies have considered static customer experiences at one point in time, whereas others have addressed dynamic experiences evolving during a longer customer journey [11]. Despite this variation in how the phenomenon should be approached, there is a consensus that customer experience refers to a customer's complex subjective reaction to offering-related stimuli [12]. In this study, we particularly implement the widely established multidimensional conceptualization of customer experience [12,14,28] to comprehensively understand different facets of the customer experience of CE offerings. The model suggests that customer experience comprises a customer's sensory, affective, behavioral, cognitive, and social responses to offering-related stimuli (Figure 1). Next, we discuss each dimension in detail.

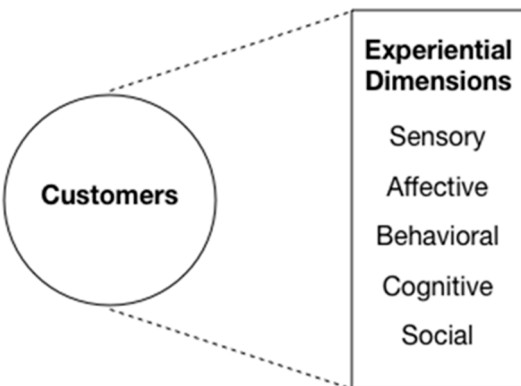

**Figure 1.** Experiential dimensions.

### 2.1.1. Sensory Experience

The sensory experience refers to a customer's experience that emerges through the five human senses: sight, sound, touch, taste, and smell [24]. Sensorial experiences typically emerge when a customer senses visual aesthetics or is otherwise sensorily stimulated by a service, product, or brand. Customers do not sense the products alone: the retailing atmosphere can also play an important role, as customers perceive it with their senses during the shopping process. For example, Douce and Janssens [35] revealed that a pleasant fragrance in a retail space positively affects customers' affective reactions, evaluations, and intentions to revisit the store in fashion retailing. Additionally, music, color, lighting, and crowd can also influence customers' shopping behavior [36].

### 2.1.2. Affective Experience

Affective experience refers to a customers' feelings and emotions that are evoked when they interact with different parts of a brand, service, or product [28]. These emotions can range from positive to negative. Virtually, firms pursue to create positive emotions, such as joy, delight, happiness, and excitement [37] for their customers, as such emotions are related to customer satisfaction, buying process, and loyalty [15,31]. However, despite the best efforts made to avoid such occasions, interactions can also evoke negative emotions, such as disappointment, frustration, unpleasantness, and dislike [37], which in turn can have negative and possibly long-lasting consequences.

### 2.1.3. Cognitive Experience

A cognitive experience emerges through creative engagement with customers' thinking or conscious mental processes [14]. It is manifested through surprising, intriguing, and provoking customers' cognitive ability. According to Blackwell et al. [38], a customer's cognition is formed through direct interaction with the offerings (services, products, and retail environment), processing secondary sources of information (word-of-mouth and online information like ads, blogs, and reviews), and comparing information against previous memories.

According to Holbrook and Hirschman [25], cognitive experience is also subconscious and personal in nature. This entails fantasies, imagery, memories, subconscious thoughts, and unconscious processes that occur during the shopping experience. Other researchers have classified cognitive experience as outcome focus, learning, think/intellectual experience, involvement, efficiency, product-quality experience, and security [39,40].

### 2.1.4. Behavioral Experience

Behavioral experience refers to behavioral actions and responses that often become apparent through lifestyle, interaction, and objects [28]. It occurs when a customer's system of values and beliefs resonates with what the brand embodies [14]. In terms of clothing brand experience, behavioral experience is best characterized by a consumer's behavioral loyalty, attitudinal attachment, consideration set, and premium price [41].

### 2.1.5. Social Experience

Social experience originates from customers relating themselves to a reference group, culture or a lifestyle [14]. Thus, in other words, social responses refer to how customers relate to others and their social environment [12,14]. These social experiences have been shown to contribute to customer satisfaction and loyalty [42]. The social dimension of consumption is also a way for consumers to produce and reproduce their social identity [14]. For example, customers can convey their personal values through visiting certain stores or purchasing particular products and services [43]. Rintamäki et al. [44] observed that consumers purchase products that exceed their budget in order to elevate their status or self-esteem. Status enhancement communicates one's social position or membership to others, and self-esteem enhancement is the transference of identity from the brand, store,

products, and other customers to the buyer [44]. Therefore, both contribute to the social dimension.

To summarize briefly, we use these five experiential dimensions as a theoretical structure while empirically studying how customers experience recycled and reused CE products. Next, we review the literature on CE business models of recycling and reuse.

### 2.2. CE Business Models of Recycling and Reuse

In CE, the production flow is recirculated through being reused, recycled, or at least reduced to minimize waste. The 2008 Waste Framework was introduced in EU legislation, and prevention, preparation for reuse, recycling, recovery, and landfill were clearly distinguished on a priority scale [45]. Lüdeke-Freund et al. [17] posited that each step and R principle in the waste hierarchy could form a business model.

Reuse and redistribution business models reintroduce used products, components, materials, or wastes as production inputs. Meanwhile, recycling business models convert wastes into products. Regarding services, companies can offer take-back management systems to collect the used or soon-to-be recycled products [17].

The primary value propositions for the customers of the reuse and redistribute business model are cheaper prices and easier access to familiar products. Thus, a major target audience for this model is cost-conscious customers [17]. A service provider or supplier creates value through the logistics of collecting and distributing the used goods and a slight enhancement of cleaning and repairing small defects. Consequently, the seller obtains a substitute for a new product and material. The value delivery process can originate from a retailer offering used products to consumers at a discounted price or directly from one consumer providing to another. However, some indirect costs might be involved in the form of commissions to be paid to the original owner or platform. Additionally, logistics costs exist, as the take-back system is required to collect and recirculate the used products [17].

The primary value proposition of recycling business models is either recycling of green inputs offered by waste collectors/processors or products obtained from manufacturers of recycled inputs [17]. Similarly, postconsumer wastes can flow back to the manufacturers to become partly, or wholly, new products. To create value, the recycling business needs specific knowledge in product design and material science. Thus, the business can process the precise physical and chemical properties of the various materials. The value capture in the recycling business model lies in the sales of recycled materials and products. The effects of recycling can influence cost reduction.

When we look at recycling and reused products from a customer perspective, we can see the increasing customer acceptance and adoption of recycled and reused fashion [9]. This is also represented in the growing interest in the slow fashion and collaborative fashion movement [46,47]. This is especially significant among female consumers, who participate in renting and swapping used clothes. According to Shrivastava et al. [34], social media influencers' endorsement of second-hand clothes drives the adoption of online platforms for used-clothes retailing.

## 3. Methods

### 3.1. Research Strategy and Data Collection

We adopted the established qualitative research strategy and an interview method to tackle the subjective and multidimensional nature of customer experience [37]. We used a multi-case approach [48] because customer experiences are explored among consumer-customers of reused, redistributed, and recycled clothing business cases. Hence, the empirical data comprised 16 consumer-customer interviews of two case companies, one based on reused clothes and the other based on recycled textiles. As we aimed to explore the various dimensions of customer experience, direct semi-structured interviews were a suitable approach to collect the relevant qualitative data on each dimension.

Suitable cases were preliminarily explored through an online search for fitting companies and discussions with researchers in the CE literature field. Consequently, multiple potential case companies for both recycling and reuse were identified. These companies were contacted for permission to access their customers. The reuse-clothes company is a privately owned non-profit non-governmental retail organization in the reuse and resale sector of consumer textiles (UFF), which resells the textile donations of private citizens across the whole of Finland. The recycled-textile company (Pure Waste) specializes in mechanical textile recycling and produces and sells casualwear clothes made entirely from recycled cotton and polyester. As a manufacturer, it primarily caters to business customers but also has a dedicated channel for market consumers. It is committed to only using materials that would otherwise go to waste.

Data gathering was conducted face-to-face among 16 customers of the case companies during May and June 2020. The interviewers were all conducted by one of the authors. The customers were approached in a shopping mall in a Northern European city. Because the customers were approached nearby the stores without appointment, they did not receive buffer time to prepare for the questions. The "at-site" interviewing allowed the interviewer to capture the fresh customer experience by the interviewees and provided rich, intense data on how they had experienced the circular product and shopping it. The drawback was that the interviews could not exceed thirty minutes' duration.

The interviewees' ages ranged between 20 and 43 years, and the gender count was divided somewhat equally between males and females. While the gender distribution was intentional to cover the input from both genders, the interviewees' age was not controlled. The detailed background information is provided in Table 1.

**Table 1.** Overview of interviewees' background.

| Interview Group | Interviewer ID | Age | Gender | Nationality | Date |
|---|---|---|---|---|---|
| Reused clothes customer group | U1 | 25 | Female | Indian | 29 May 2020 |
| | U2 | 27 | Female | Finnish | 29 May 2020 |
| | U3 | 20 | Female | Finnish | 3 June 2020 |
| | U4 | 36 | Male | Finnish | 3 June 2020 |
| | U5 | 36 | Male | Moroccan | 3 June 2020 |
| | U6 | 27 | Female | Finnish | 3 June 2020 |
| | U7 | 28 | Male | Finnish | 3 June 2020 |
| | U8 | 36 | Male | Finnish | 4 June 2020 |
| Recycled clothes customer group | P1 | 40 | Male | Finnish | 26 June 2020 |
| | P2 | 21 | Female | Finnish | 26 June 2020 |
| | P3 | 43 | Male | Unknown | 26 June 2020 |
| | P4 | 24 | Female | Finnish | 26 June 2020 |
| | P5 | 28 | Male | Finnish | 26 June 2020 |
| | P6 | 26 | Male | Finnish | 26 June 2020 |
| | P7 | 32 | Female | Finnish | 26 June 2020 |
| | P8 | 28 | Male | Finnish | 26 June 2020 |

The interviews were semi-structured and based on the structure of an established experience dimension model. The questions covered the five experiential dimensions (sensory, affective, cognitive, behavioral, and social) regarding clothing purchase. For example, the following questions from the interview guide were posed to examine the five dimensions: "*How do you perceive this brand's product visually?*" (sensory), "*How do you feel about owning and using this brand's product?*" (affective), "*Did you learn anything new when owning/using this brand's product?*" (cognitive), "*Does owning/using this product have any impact on your day-to-day activities?*" (behavioral), "*How do your friends and family feel when they see you owning/using this product?*" (social). Additional questions were asked to allow interviewees to elaborate on their points, which encouraged exploration. The interviews were recorded and transcribed for analysis.

As our research design was qualitative and explorative, the data set of 16 intensive, yet content-rich interviews were considered an appropriate amount of data for examining experience dimensions and how they manifest. In qualitative studies on customer experiences, the number of informants has been even fewer than 10 interviews [49]. Here the data collection and amount of data was also guided by saturation, which refers to continued sampling until no new substantive information is collected [50]. The saturation point was reached at the sixth interviewee from the reused clothes group and seventh interviewee from the recycled clothes group. However, at least one more interview was conducted to ensure data saturation was achieved. For example, when inquiring the interviewees' sensory experience on recycled clothes, most of the factors regarding style, aesthetics, and comfort were mentioned already in the first few interviews.

### 3.2. Data Analysis

The analysis followed the established procedure of qualitative content analysis and thematic analysis. The interviews were first reviewed individually and then cross-analyzed to examine the common themes.

A comprehensive coding category system was developed to summarize the collected answers. First, the transcription of the interviews was put as 16 unique documents into Atlas.ti, a qualitative data analysis and research software. Second, these parts were segmented into coding units using Atlas.ti. The coding framework was then developed based on a thorough review of the transcription. Twenty-one unique codes were created—partly deductively, based on the theoretical experiential dimensions, and partly inductively, based on an explorative, grounded theory approach. Each coding unit was a piece of information or idea that was understandable by itself, such as aesthetics, colors, basic, sustainable, and water. These coding units were assigned to transcription sentences that indicated the code's meaning. For example, if a sentence talks about whether the interviewee found the clothes beautiful, then it would be labeled under the code "aesthetics." The predetermined codes served as superior codes that would include the codes that labeled the sentences. For example, the superior code "sensory" would include words such as "aesthetics" and "colors". Initially, some coding units were categorized using just one word, but it did not illustrate a noteworthy concept well enough. For example, the word "think" might be a cue for a cognitive experience unit code, but some people use it habitually in a sentence without necessarily indicating a cognitive process occurrence. Consequently, all coding units in this study were sentence-long.

Each customer experience dimension had a subcategory, where appropriate coding units were placed using induction. For example, the sensory dimension includes the five human senses; the visual sense can experience colors and beauty, and the sense of touch can experience the softness and comfort of the textile. Thus, the codes for "colors," "aesthetics," "softness," and "comfort" were put under the "sensory" subcategory.

The quality of the results of this explorative study was ensured through multiple means; for example, the analysis program and structured analysis procedure ensured the rigor of the analysis. Furthermore, saturation was reached, as explained in Section 3.1.

## 4. Results: Customer Experience Dimensions of Reused and Recycled Clothes

Next, we discuss our findings on how customer perceive the CE offerings regarding the five experiential dimensions of customer experience for reused and recycled clothes. We provide our results for each experiential dimension on both reused and recycled clothes along with empirical examples.

### 4.1. Sensory Experience

In the case of clothing brands, the sensory experience was divided into three categories, which are detailed in Table 2.

**Table 2.** Sensory experience of reused clothes and recycled textiles among the interviewees.

| Sensory Experience: Sensing Circulated Materials and Products Through | Experience Remarks from Reused-Clothes Interviewees | Experience Remarks from Recycled-Textile Interviewees |
|---|---|---|
| Sight | Interviewees liked the clothes' style, found the products in the store aesthetically pleasing, and noted that the products were in good condition, without any damage. | Interviewees preferred the simple color scheme, liked the basic look (without any big logo), and appreciated the products aesthetics. |
| Smell | No interviewee had issues with the smell of the used clothes. | No interviewee had any particular issue with the smell. They found no strong chemical smell that new clothes usually have. |
| Touch | Interviewees liked reused clothes that were soft and comfortable to wear. | Interviewees found that the clothes made from recycled materials were soft and comfortable. |

Regarding sensory experiences originating from reused and recycled clothes, most remarks were related to sight, smell, and touch. Two senses, sound and taste of circulated clothes, were not mentioned by any interviewee, which is understandable because clothing and textile have no significant characteristics regarding these two senses.

Regarding the sense of sight, experiences originated from three main characteristics of the products: color, style, and aesthetics. The interviewees cited color as one of the most important factors when they considered buying a circulated shirt. As the garments are used for a longer period, the owners appreciate when their color is retained. Regarding the style of garments, having the style to fit the customer was mentioned as a crucial factor. Interviewees from the recycled-textile store preferred the basic style with simple color, which is characteristic of the brand. In contrast, reused-clothes customers gravitated toward stylish and "cool"-looking clothes. A few interviewees from both brands focused on the overall aesthetics of the circulated products. They explicitly mentioned that the garments should look good on them. The reused-clothes interviewees also focused on the condition of the products, as the products would not be brand new.

Smell was frequently mentioned by most interviewees, but it was unanimously agreed that there was nothing particularly negative. Although the products were used, the reused-clothes customers did not have any problem regarding the smell. Furthermore, they gave smell less significance, as they explained that they can always wash the clothes to get rid of the smell. Meanwhile, one interviewee observed that, while new products had a strong chemical smell, the recycled textile did not.

The sense of touch was strongly commented on by both the recycled-textile and reused-clothes customers, making it another major positive sensory characteristic. The comfort and softness of the recycled textile received multiple praises. For example, the brand won one interviewee's loyalty to its hoodies because of how comfortable the clothes were. Reused-clothes customers also noted comfort and softness as critical factors: reused clothes needed to feel comfortable when moving one's arms and legs.

*4.2. Affective Experience*

The feelings of affective experience for customers of both companies were largely positive and are elaborated in Table 3.

**Table 3.** Affective experience of the interviewees.

| Affective Experience | Experience Remarks from Reused-Clothes Interviewees | Experience Remarks from Recycled-Textile-Clothes Interviewees |
|---|---|---|
| <ul><li>Positive emotions on circulated garments and materials</li><li>Positive emotions on sustainable dressing</li><li>Positive emotions on the brand enabling textile circulation and sustainable dressing</li><li>Positive emotions on economic benefits and good price/quality/impact ratio</li><li>Negative and neutral emotions (e.g., shame and curiosity)</li></ul> | Interviewees felt positive about the sensory experience and buying used clothes.<br><br>They are good for sustainability reasons, for paying cheap prices, and when talking to social circles about their reused-clothes purchases. Interviewees considered the items as only utilities. | Interviewees felt positive about buying recycled clothes, the brand's recycling process, and the sensory experience.<br><br>Interviewees also liked used clothes.<br><br>Interviewees felt good about representing a sustainable brand |

The positive emotions of customers were reported for the circulated garment, the brand enabling sustainable fashion, sustainable textile material, and sustainable dressing per se. When talking about affective experience toward the reused-clothes company, a couple of interviewees derived positive emotions from the pleasantness of the visual of the clothes. They also felt positive about buying used clothes rather than new clothes. Several interviewees appreciated the fact that the material already existed and no further natural resources needed to be used. This appreciation originates from their shared belief in ecological values. Furthermore, positive feelings occurred when beautiful or unique clothes were found. An interviewee, for example, felt trust in the brand's ability to carefully pick its products and maintain a certain quality, whereas another interviewee expressed a positive experience every time they purchased from the reused-clothes company. Another source of positive feelings arose from purchasing cheaper clothes.

By contrast, several interviewees conveyed a lack of emotional attachment to the reused-clothes company's products. They treated clothes as a utility, basing their value on their quality alone. They were primarily concerned about the condition and price of the products. Interestingly, one interviewee mentioned that she/he did not perceive products sold in the reused-clothes company stores as their products, but rather the brand from which the products originated.

For the recycled-textile customers, the affective experience lied in the fact that money was spent on an ethical brand. Interviewees explained that wearing unethical clothes made them feel conscious. They also felt the pressure of other people judging if they were not wearing ethical clothes. By owning ethical clothes, they could overcome the guilt and the perceived social pressure. Other recycled-textile interviewees felt good just from the fact that the clothes were recycled. Using "something sustainable and recycled, rather than something like fast-fashion" created a better feeling. The interviewees also enjoyed the sensory attributes of the clothes, such as simplicity, basic style, color, and comfort. For example, one interviewee stated that their feeling toward the recycled-textile company was "warmish", which was more than they would usually have for clothes.

### 4.3. Cognitive Experience

Both interviewee groups witnessed several cognitive experiences, which are detailed in Table 4.

The recycled-textile interviewees learnt more about the process of how the company makes its fully recycled garments. For example, the brand's name sparked curiosity for one interviewee and led them to visit its website, where they learned about the processes, and another was educated about how to extend the clothes' lifecycle. Another identified aspect was how cognitions and values were linked: one recycled-textile interviewee noted that they consciously tied their values with their clothing purchase decision, and if they did not wear sustainable clothes, they would feel conscious.

Table 4. Cognitive experiences of interviewees.

| Cognitive Experience: | Experience Remarks from Reused-Clothes Interviewees | Experience Remarks from Recycled-Textile Interviewees |
| --- | --- | --- |
| • Learning on the textile processing needed for clothes, and the link between own behavior and environmental impact<br>• Comparing the consequences of using "virgin" vs. circulated clothes<br>• Being skeptical toward sustainability of circular clothing brands | Interviewees did not view reused clothes as second-hand,<br>changed opinion about the use of second-hand clothes,<br>realized that it was not necessary to consume a lot,<br>understood the positive environmental impact of buying used clothes,<br>realized how cheap the prices were in second-hand stores,<br>distinguished the different brands of clothes in the store,<br>and were skeptical of the brand. | Interviewees learned how to prolong the clothing lifecycle,<br>learned about the manufacturing process,<br><br>connected personal values and other factors with purchasing decisions,<br>transferred their knowledge to others,<br>compared the company with other brands,<br>and were skeptical of the brand. |

The cognitive experience was present in conjunction with social experience: the interviewees educated their colleagues and friends about the recycled-textile company's recycling concept. Knowledge and information about ethical textile manufacturing processes were transferred from recycled-textile interviewees to their colleagues.

The cognitive experience occurred as the customers made their purchase decisions. In the case of physical products, such as textile products, the cognitive experience was related to sensory experience. In particular, sight and touch, as well as smell, served as inputs for the customers to process their rationale for buying from the brands. For example, the comfort of hoodies made from recycled textiles persuaded customers to commit to the purchase.

Findings concerning the cognition dimension indicated that customers also unlearned prejudices concerning circulated clothes and their minor quality or attractiveness: for example, interestingly, one reused-clothes interviewee did not think of the reused-clothes company's products as used clothes. The quality of these clothes exceeded their expectations, making them feel like they were not second-hand clothes. Another interviewee had a mindset shift about buying used clothes. They originally came from India, where wearing used clothes is unusual. Now they are comfortable buying second-hand clothes because this practice is normalized in Finland and is good for the environment. Another interviewee was inspired by the purchase of used clothes and started thinking of new ways to use less in other aspects of everyday life. They wanted to minimize their daily output, as they realized that they did not need as much as they thought. They also understood the positive environmental impact that resulted from buying from the reused-clothes company and other second-hand shops. Two other interviewees also agreed with the sustainability that resulted from purchasing second-hand clothes. One reused-clothes interviewee explained how surprised he was by how cheap certain items were in second-hand stores. The cognitive experience was highlighted in another reused-clothes customer's shifted perception of used clothes, as the quality of most purchased products surpassed their expectations.

One reused-clothes interviewee made a logical disconnection between the brand of the reused-clothes company and the products in the store, as they originally came from somewhere else. The interviewee compared the recycled-textile company's products and the fast-fashion industry's products based on their environmental impacts and garment quality. As the company only used recycled material, the interviewees deemed it better for the environment than fast-fashion brands. Furthermore, its products stood out compared to cheap fast-fashion clothes, which did not retain their form after washing. Several interviewees stated that their learning on sustainable clothes did not come from clothing brand communication; instead, knowledge and information about the environment were accumulated through documentaries and media.

Another noteworthy finding was that customers perceive skepticism in their cognitive processes concerning circulated clothes; for example, one interviewee was skeptical toward the brand's sustainable PR because they felt that most of the positivity came from the recycled-textile company itself. Similarly, the reused-clothes interviewee thought that the reused-clothes company in question was "shady" as, although it is a non-profit organization, some of its leading managers have committed financial crimes. Several recycled-textile interviewees expressed their interest in second-hand shops. Purchasing from these shops was in alignment with their sustainability values, even though determining the origin of these garments might be harder. Nonetheless, they still preferred buying new garments.

### 4.4. Behavioral Experience

The behavioral experience reported by the interviewees revolved around the changes in the way they approached sustainable consumption. These changes are described in Table 5.

**Table 5.** Behavioral experience of the interviewees.

| **Behavioral Experience:** | **Experience Remarks from Reused-Clothes Interviewees** | **Experience Remarks from Recycled-Textile Interviewees** |
|---|---|---|
| • New practices and buying behavior in buying frequency (buying less) and marketplaces (more second-hand) <br> • Implementing sustainable decision-making criteria (sustainability and impact, ethicality, price-quality-impact ratio) | Interviewees bought fewer new clothes, <br><br> went to other second-hand stores and virtual marketplaces, <br><br> bought used items for others, and went to the reused-clothe-company's store for specific items. | Interviewees bought only sustainable and ethical clothes, <br><br> became more loyal to the brand, <br><br> taught others about the brand and sustainability, bought used items for others, <br> researched the brand and sustainability of products. |

Regarding the behavioral dimension, the customers of reused clothes and recycled textile increasingly implemented practices and decision-making criteria in which sustainability played some role, even though they did not consider their actual behavior to change.

The main behavioral findings concerned consumers' new practices and buying behavior in terms of buying frequency and the marketplaces they turned to when requiring "new" clothes, as well as the decision-making criteria they implemented. Several reused-clothes interviewees had a shift in their purchase behavior after buying used clothes. They rarely bought new clothes and tried to consume less. This translated to consumption in other areas as well, such as furniture and toys. They also went to other second-hand shops or flea markets, besides the current reused-clothes company. Thus, sustainability played a major decision-making role in behavior. To give examples, one recycled-textile interviewee's purchase behavior was influenced by the sustainability value. The interviewee explained that, as she worked in the Finnish environment institute, she and the colleagues often made sure that their outfits were sustainably produced. Similarly, another recycled-textile interviewee stated that buying ethical clothes was "the new normal" for them. Purchasing clothes that are neither second-hand nor recycled was no longer an option. A few interviewees experienced buying and consuming circulated clothes as ambivalent behavior: one argued that the best behavior would be not to buy anything at all, but agreed that it was good to support stores that are trying to make a difference.

Positive experiences with circulated clothes also led to loyal behavior; for example, the recycled-textile company's impressive hoodies' production persuaded some customers to be loyal customers: of all the hoodies that they had tried, these exceeded every other brand.

The behavioral experience also exhibited itself in the social and emotional dimensions. The reused-clothes' interviewees often shared proudly interesting items that they purchased in the store for a cheap price. This was done with the intention of surprising the parents and friends. One interviewee explained they would buy cheap used items from the

reused-clothes company and give them away to others. In contrast, the recycled-textile interviewees forwarded the information they learnt about the recycled-textile company's recycling concept and sustainability issues to their social network. Thus, their behavioral experience was manifested in the act of educating themselves and others.

*4.5. Social Experience*

In this study, social experiences are centered on the relationship between customers of circulated clothes and their referenced social group. The main points are summarized in Table 6.

**Table 6.** Social experience of the interviewees.

| Social Experience: | Experience Remarks from Reused-Clothes Interviewees | Experience Remarks from Recycled-Textile Interviewees |
|---|---|---|
| • Gaining positive feedback from social networks<br>• Educational interaction<br>• Manifesting personal values<br>• Making an impact in social network | Interviewees received positive feedback from family and friends in Finland,<br><br>received questions from relatives of different cultures,<br><br>wanted to prove the quality and cheapness of used clothes and believed they are making a positive impact. | Interviewees' friends and family do not often notice that the clothes are recycled material, but those who did, express curiosity and interest. Interviewees supported ethical, environmentally sustainable brands and<br><br>hoped that they are bringing about a positive impact. |

The main findings of the social dimension of customer experience, originating from circulated clothes, were perceived positive feedback from social networks, educational interaction about sustainable clothing with friends, manifestation of sustainable values for others, and making an impact on social networks.

Regarding the social interaction and feedback triggered by circulated textiles, most of the interviewees (Europeans) reported that they gained positive feedback from their social networks when discussing their circulated clothing purchases. However, this positiveness of social experience might be cultural-context-specific as one interviewee who originated another regional context (India) reported the opposite social experience: As Indian culture is not familiar with wearing other people's clothes, the interviewee's Indian relatives questioned their choice of buying second-hand clothes.

In general, the customers of the recycled-textile company received curiosity and positivity from their friends about their clothes being made of recycled fibers. This also led to educational discussions and interaction in the customers' social networks: a few interviewees stated that they discussed the brand and sustainability with their colleagues at work.

The interviewed customers also stated that they were causing a societal impact through the act of purchasing products from either of the case companies. They experienced that reused and recycled-material-made clothes enabled them to manifest their identity and values: several wanted to prove to their friends and family that quality garments can come cheap and wanted to showcase the good ratio among quality, price, and environmental benefits in their social network. The customers also argued that they wanted to support ethical brands like the recycled-textile company because they offer fair wages to the workers.

**5. Discussion: Developing a Synthesis of Customer Experience Dimensions of Circulated Clothes**

Our exploratory analysis of the five dimensions of customer experience allows us to synthesize how the customer experience of circulated clothes (from reuse and recycling business models) is constituted. The results of the empirical study and the synthesis enhance the understanding of the customers' experience of the circulated products of the clothing industry and of customers' motivation to use recycled and reused products and adopt CE products. The contents of the five experience dimensions (Figure 2) explain what

constitutes customer experience in the sustainable clothing industry. For each dimension, numerous experience elements were observed and conceptualized: for the sensory dimension, touch, smell, and aesthetic look were the most important; for the affective dimension, positive emotions, such as pride, surprise, and satisfaction, shaped emotions; for the cognitive dimension, learning about sustainable clothes and impacts was important; for the behavioral dimension, new knowledge to be implemented into new practices, including renewed buying behavior in terms of buying frequency, criteria, and loyalty, was important; and for the social dimension, interactions and discussions on sustainable clothing, positive feedback, and manifestations of identity and values were important.

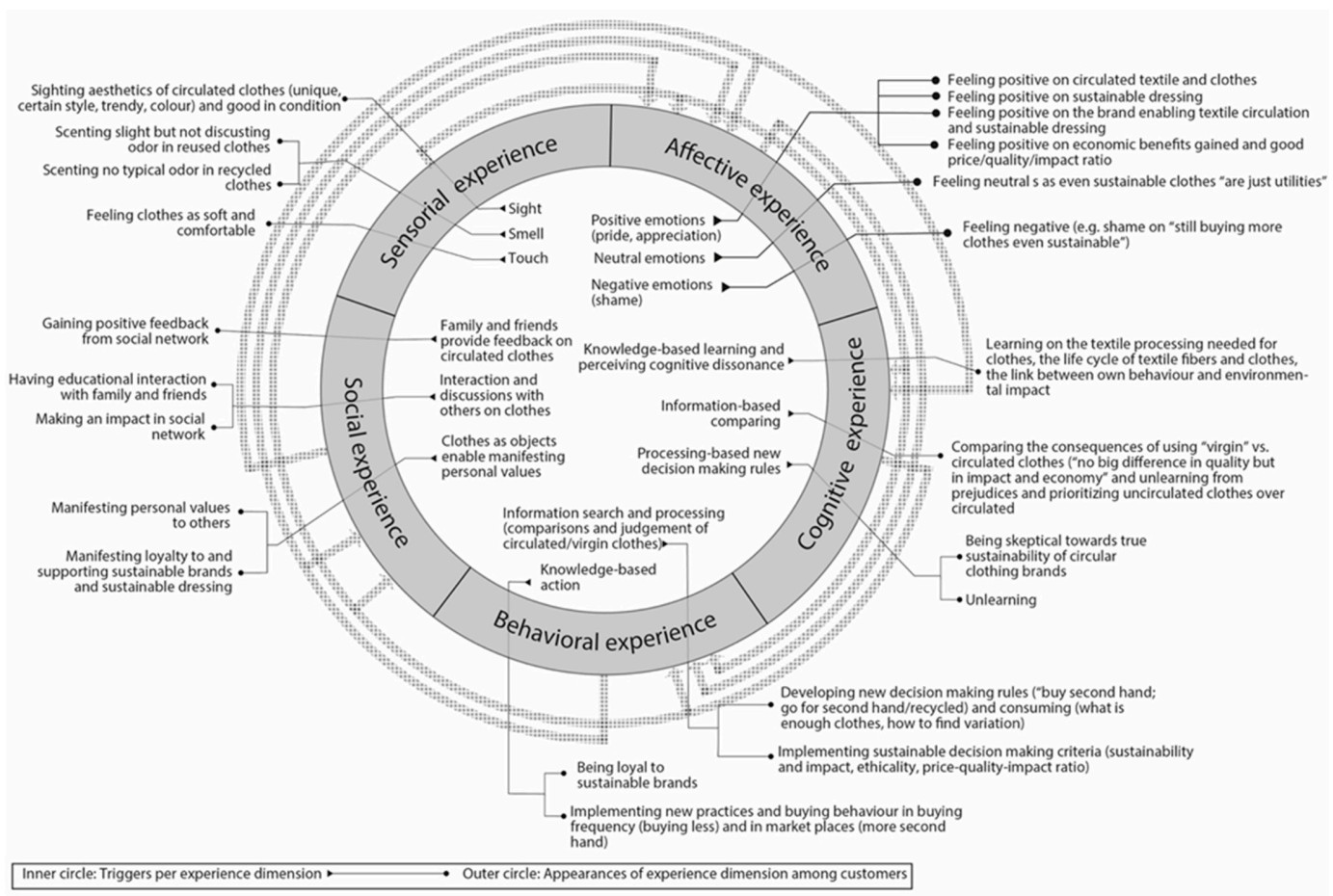

**Figure 2.** Empirical-based model on customer experience dimensions in circular economy products: how consumer-customers experience recycled-textile and reused clothes.

Cognitive experience is a dimension that includes numerous insights. The customers experienced cognitive discoveries, which occurred as the customers experienced the clothes quality in their daily lives and learned a good price–quality ratio, and environmental benefits blended into renewed decision-making criteria as a result of learning and unlearning. The case companies whose customers we interviewed also actively engaged customers in creating cognitive experiences and thereby learning about the manufacturing process and the ecological and ethical impacts of sustainable clothes.

In the behavioral dimension, the interviewees exhibited switching behaviors, as they had a declining interest in unsustainable clothes and increasing interest in used and recycled clothes. The interviewees frequently visited different second-hand stores and marketplaces. An evidence of this switching behavior was presented by Wu et al. [51]. Additionally, indications of philanthropy behaviors were found when the interviewees purchased used items for not only themselves but also other people.

The social dimension shows how customer experience regarding circulated products is characterized by interaction with others and their acknowledgment for circular choice, as well as customers' ability to convey and manifest their personal, sustainable values to their reference group. As Gupta and Ogden [52] found evidence of trust, in-group identity, and perceived efficacy in sustainable customers, this study of both groups' interviewees also exhibited similar evidence. The in-group identity, or one's identification with sustainable consumption, was present among customers who highlighted strong value alignment between themselves and sustainability.

Regarding the affective dimension, customers experienced diverse positive and negative emotions such as shame, pride, compassion, admiration, confidence, and trust, which resulted in and from the social dimension. For example, the positive feedback from friends and family fueled customers' positive emotions.

To summarize our key findings, we first itemized how diverse experiential dimensions are manifested and thus, revealed the diverse experiential aspects that constitute the customer experience of recycled and reused circulated clothes. Second, we developed an initial understanding of how the different experiential dimensions interact, intertwine, and accumulate. Our analysis revealed some major interconnections between the dimensions; for example, consumers felt pressure from other people's judgement of unsustainable clothes, which led to feelings of shame or joy, pride, and satisfaction. These feelings then translated into new decision-making criteria and buying practices under the behavioral dimension and interaction and manifestations under the social dimension. Similarly, as customers learned more about sustainable clothing (cognitive dimension), this led to educational discussions with friends and colleagues (social dimension). The findings originating from CE customer experiences were aligned with customer experience research arguing that the experiences were dynamic, partly due to interconnected stimulus [12]: our study mapped diverse elements from the five experiential dimensions of CE experiences and captured their interactions as well.

## 6. Conclusions

### 6.1. Theoretical Contributions

This study examined how customers experience CE products, particularly circulated clothing; it structurally mapped the customer experience dimensions of the reuse and redistribute business model and recycle business model in the clothing industry.

The study findings make several contributions to the CE and to circular business research. This study is the first to use customer experience lenses and an experiential dimension model to study CE business. This approach allows this interview study among customers to develop a new understanding of how customers experience CE products—here recycled textile and reused clothes—and how customers' experiences of CE products are based on how they sense, feel, know, act, and socially interact. These findings provide a new and seldom-studied microlevel perspective [21–23] to CE business model research, which more often focuses on different business model types and company businesses [4]. Our findings on how customers experience circulated clothes through the five experiential dimensions (sensing, feeling, knowing, acting, and socially interacting) extend the understanding of CE business from a customer's perspective, particularly in the fashion, textile, and clothing industries. Thus, our findings also contribute to the studies examining CE actions in these sectors [5,7,8,23]. Our findings also contribute to previous discussions on customers' adoption of reused products, particularly in the fashion and textile industries [9], as well as green consumerism [53]. In summary, this study builds a crucial micro-consumer-level understanding of CE, which is an important contribution to the literature, as customer perspective in CE research has been neglected so far, under the dominance of business models and other perspectives.

Furthermore, our study contributes secondarily to the customer experience and experience marketing pioneered by Schmitt [28] and Brakus et al. [27], as well as green consumerism research, as we have used the established experience dimension approach in

novel, sustainable and circular product settings. Our results explain how circular products are experienced per dimension, and the proposed empirical-based dimensional model displays what constitutes CE customer experiences. These insights build a new understanding on how customers experience circular products—the aspect that is missing from the extant linear-product oriented customer experience research. We investigated how the five experiential dimensions—sensory, affective, cognitive, behavioral, and social—appear when consumers experience sustainable, circulated products. Our study findings also provide interesting empirical insights into how the experiential dimensions of CE are interlinked and interact (Figure 2). These findings partly agree with previous studies, arguing that, for example, affective experience is related to sensory experience [35,54]. Furthermore, in the environmental context, Hartmann et al. [55] theorized the occurrence of auto-expression benefits, which were also found in this study, as most interviewees felt good about their contribution to the environment through sustainable purchases and manifestation of their sustainability values. There were ample cognitive experiences through customers' interactions with the stores' offerings, secondary sources of information, and previous memories, which agree with previously reported results [38]. These cognitive experiences form learnings, intellects, and purchase outcomes [39,40]. Such learning relates to the quality and utility of used clothing and the manufacturing process of recycled clothes. According to Schmitt [28], behavioral experience occurs when customers gain a refreshing way to view lifestyle, interaction, and objects. In this study, such behavioral experience was observed in the form of interviewees' changing behavior. However, it did not originate externally from influencers, as Schmitt [28] theorized, but rather through an internal cognitive process. This study uncovered, for example, positive social feedback obtained from friends and family and a sense of pride in increasing sustainability and impact. This yields evidence of social emotions [56]. Furthermore, buying circulated products is a means and object which communicate personal values to others [43], and customers manifest their loyalty for sustainable, ethical brands [42].

Our study displayed the interaction of the experiential dimensions: how, for example, learning about textile circulation and environmental impact (cognition) leads to positive and negative emotions (such as shame and pride). These findings indicate that how customers experience circulated products is a dynamic phenomenon in which knowledge, emotions, and social interaction are related, all of which together shape customers' willingness to adopt, choose, and buy circulated products.

Interestingly, our findings also indicate that diverse experience dimensions together may build the base for ethical judgement for the more circular choice: our data indicated the affective, cognitive, and social experience dimensions (feeling positive on circular choice; learning sustainability choice criteria; getting positive feedback on circular choice) together support or constitute ethical judgment [57] towards the circular product choice.

*6.2. Managerial Implications*

Our findings also provide pragmatic guidance for managers in companies that operate on the basis of the reuse and redistribute business model and the recycle business model and those particularly in the clothing business. To holistically improve the customer experience of circulated products, all dimensions should be acknowledged. Managers with recycled material and reused products should acknowledge the sensory marketing aspect and provide experience triggers through not only the visual sense but all human senses to make their sustainable offerings intriguing. The positive affective experience that arises after the purchase is often for sustainable and ethical reasons, even though what initially brought them to the store might be egoistic reasons, such as a cheaper price. Therefore, stores that sell reused clothes should focus on showing the sustainability impact, as according to the waste hierarchy, reuse is the more preferred way of minimizing waste than recycling. This can provide customers with another reason to buy, which is a positive emotion originating from sustainable purchases. Third, the cognitive dimension witnesses a change in the interviewees' perception of the quality and sustainability of used clothes, and therefore,

raising awareness about these aspects of used products is crucial. Several recommendations concern the social dimension: managers can highlight the societal benefits that customers make when they purchase circulated products. This can be done by creating a community wherein customers celebrate the positive contribution to sustainability, are proud of their purchase of used and recycled clothes, and would like to share their purchases with friends and families. This suggests sustainable brands emphasize the perceived status reward of owning and using the products. Finally, as customers teach others about the brand and sustainability, managers should leverage this observation to spread their awareness of both the brand and sustainability.

*6.3. Limitations and Future Research*

This qualitative exploratory interview study also has limitations. We interviewed a limited set of customers in one geographical location, Northern Europe, which limits the transferability of the results. In addition, data collection in qualitative research is often unstructured and subjectively interpreted. Furthermore, the interviews were conducted in English but in Finland. As a result, some terminologies and sentences might have been lost in translation or omitted entirely. Second, the interviewer's verbal or non-verbal communication during the interviews could have created bias. Furthermore, our case sampling was limited, as there is only one case company representing each business model. The collected data might not be fully representative of the whole spectrum of customer experience in each business model. Herein, we studied customer experience of circulated products in the clothing, textile, and fashion industries. Thus, more case studies need to be conducted in different locations and industries/sectors/product categories to validate the current research and broaden the empirical results.

Despite these limitations, this study opens several pathways to explore relevant future research in the fields of customer experience and CE. This study emphasizes the identification and rationalization of the customer experiential dimensions and values in the reuse and redistribute business model and the recycle business model in the clothing industry. Moreover, the management of these experiences and values to enhance the CE ecosystem remains largely unaddressed. As our emerging findings suggest that experiential dimensions interact and even accumulate, it would be valuable to study in a more focused manner how dimensions interact when customers experience circular products. Future studies could also examine in more detail the diverse touchpoints, stimuli, and moderators that shape customers' circular product experiences. Furthermore, the CE business models are not limited to only the recycle model and reuse and redistribute model, but other factors, such as sharing and renting clothes, should also be studied in the way they create the customer experience dimensions. Finally, there are interesting cultural differences in how circulated products are perceived, and therefore, cross-country comparisons can provide a better understanding of how to increase the adoption of CE products among diverse customers globally.

**Author Contributions:** Conceptualization, L.A.-S., A.H.T. and L.L.; methodology, A.H.T.; writing—original draft preparation, A.H.T., L.A.-S. and L.L.; writing—review and editing, L.A.-S., A.H.T. and L.L.; visualization, L.L. supervision, L.A.-S.; project administration, L.A.-S.; funding acquisition, L.A.-S. All authors have read and agreed to the published version of the manuscript.

**Funding:** This research was funded by the Strategic Research Council, Academy of Finland through the project entitled "Circular Economy Catalysts: From Innovation to Business Ecosystems" (CICAT2025) (grant ID 320194).

**Institutional Review Board Statement:** The study was conducted according to the guidelines of the Declaration of Helsinki and approved by Tampere University and the CICAT project.

**Informed Consent Statement:** Informed consent was obtained from all subjects involved in the study.

**Data Availability Statement:** Data available on request due to restrictions (to protect privacy of the informants). The data presented in this study are available on request from the corresponding author.

**Conflicts of Interest:** Authors declare no conflict of interest.

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
