# Peer review of "Customer Experience in Circular Economy: Experiential Dimensions among Consumers of Reused and Recycled Clothes"

_sustainability, doi:10.3390/su14010509_

Round 1

Reviewer 1 Report

Paper is well written and structured. Introduction section clearly presents paper objective and motivations. The literature analysis is connected with paper objectives. In addition, findings are interesting, and discussion section has been developed properly. Therefore, I see some merit in the actual contents of the paper.

  1. My main concern is related with the data collection. How can we know that 16 customers are enough to obtain suitable conclusions? Authors need to prove that.
  2. On the other hand, overall, try to provide sufficient validation regarding the novelty of this research along with beneficial.

Author Response

Paper is well written and structured. Introduction section clearly presents paper objective and motivations. The literature analysis is connected with paper objectives. In addition, findings are interesting, and discussion section has been developed properly. Therefore, I see some merit in the actual contents of the paper.

  • Thank you for this encouraging comment: we are pleased to hear that you see contributions in our findings and consider our paper well written. We have now put much efforts to provide the version where we have improved flaws pointed by you and the other reviewers and we hope that this version reads even better.

My main concern is related with the data collection. How can we know that 16 customers are enough to obtain suitable conclusions? Authors need to prove that.

  • We understand your concern on the amount of data but we would like to also highlight here that our approach was explorative and qualitative. the earlier published studies on customer experience have been grounded even lesser amount of data (e.g. 7 informants in paper by Dube & Helkkula, 2015). As our research goal was to explore and examine how the five experiential dimensions manifest when consumers experience circulated products (clothes). During the research process we faced saturation, meaning that no new, particularly relevant information emerged from interviews, after the seventh interview per category, regarding the dimensions. Therefore, conducting more interviews would not have significantly changed the main experiential triggers identified.
  • In the revised version we explain in more detailed way, how earlier studies and saturation point guided our data gathering and how we assessed that we have enough data to develop our results and the model.
  • The text is revised in the following way:

“As our research design was qualitative and explorative, the data set of 16 intensive, yet content-rich interviews were considered an appropriate amount of data for examining experience dimensions and how they manifest. In qualitative studies on customer experiences, the amount of interviews have been even lesser that 10 interviews (see e.g.  Dube & Helkkula, 2015). Here the data collection and amount of data was guided also by saturation, which refers to continued sampling until no new substantive information is collected [48](Miles and Huberman, 1994). The saturation point was reached at the sixth interviewee from the reused clothes group and seventh interviewee from the recycled clothes group. However, at least one more interview was conducted to ensure data saturation was achieved. For example, when inquiring the interviewees’ sensory experience on recycled clothes, most of the factors regarding style, aesthetics, and comfort were mentioned already in the first few interviews.”

On the other hand, overall, try to provide sufficient validation regarding the novelty of this research along with beneficial.

  • We definitely agree that it is important to explain clearly what the novelty of this research is. Therefore, we have developed the sections that explain our contributions. In the revised version we for example justify the novelty in a more sharp and explicit way.
  • In the introduction, there is a new paragraph explaining the existing gaps and the novelty of our study: “To summarize the extant gaps in the relevant literature, the CE and environmental sustainability research has neglected customer and particularly customer experience that would explain how circularity and circular products are experienced by customers, and customer experience research has implicitly focused to analyzing linear products and not has not therefore addressed particularly circular products, and is therefore lacking specific understanding of how customer experiences are constructed in circular products.”
  • In conclusion section, we argue our contribution in more explicit manner: “Our results explain how circular products are experienced per dimension, and the proposed empirical-based dimensional model displays what constitutes CE customer experiences, and these insights build new understanding on how customer experience circular products as this aspect is missing from the extant linear-product oriented customer experience literature.”

Reviewer 2 Report

Thus, the purpose of this research is to understand the customer perspective of recycling and reuse. The research seeks answer to pertinent question of how Circular Economy products (recycled vs. reused) experienced by consumers in the clothing industry. To find answer to their research question authors have conducted a qualitative interview study to explore how consumer-customers experience recycled textiles and reused clothes. Authors have used an established experience dimension model and mapped how the five dimensions of customer experience (sensory, affective, behavioral, cognitive, and social) present themselves in the sustainable clothing industry. The manuscript definitely provides new information which is significant but requires some improvement. My comments and suggestions are as follows:

Abstract

Abstract is well written and captures the main highlights in a very lucid manner. Only suggestion for authors is to arrange the keywords alphabetically.

  1. Introduction

Authors need to strengthen the introduction section further. I suggest authors introduce the customer perspective on recycling and reuse in general without a definition which could be provided.

On page 2 line 48 authors mention that “….customer experience has been identified as a compelling antecedent in various business contexts….” but they do not tell their readers what customer experience means and how is relevant in this specific context. Authors may want to read and cite following recently published paper that covers customer experience in detail:  https://doi.org/10.1016/j.jretconser.2021.102798

 In the later section of the introduction authors have identified and mentioned the key gaps but this must be followed by clearly stating the novelty of the present study. Study's objectives can also be clearly stated through research questions in the introduction section.

  1. Theoretical Background

This section is well written but I would suggest that on page 5 after line 207 authors may want capture the ethical judgement of customer to highlight recycle and reuse. Term ethical judgement along with religiosity for the purpose of environmental sustainability has been captured in the following paper: https://doi.org/10.1016/j.techfore.2021.121094

Authors may want to give a careful read to this section and entire manuscript for identifying typos that may be there.

  1. Methods

Methods and data analysis is the real strength of this paper. However, I have some issues that authors may want to explain and incorporate in the revised version of their manuscript.

On page 6 line 246 authors have mentioned that empirical data comprised 16 consumer-customer interviews of two case companies. Since the sample size of 16 seems too small therefore, I suggest authors provide some justification for the understanding of readers on how this is appropriate for this type of research. Also on same page line 265 and 266 authors mention that “…..the interviewees’ ages ranged between 20 and 43 years, and the gender count was divided somewhat equally between males and females…” Authors may want to provide some explanation on why the age groups and gender?

  1. Results

This section is the real strength of this paper but I suggest authors highlight the value of the paper in few lines at the end of this section. Authors may want to elaborate on this by explaining …how and why?

  1. Discussion and conclusion

These sections are nicely written and explained. I don’t see any problem with this. Wish authors all the best.

Author Response

Thus, the purpose of this research is to understand the customer perspective of recycling and reuse. The research seeks answer to pertinent question of how Circular Economy products (recycled vs. reused) experienced by consumers in the clothing industry. To find answer to their research question authors have conducted a qualitative interview study to explore how consumer-customers experience recycled textiles and reused clothes. Authors have used an established experience dimension model and mapped how the five dimensions of customer experience (sensory, affective, behavioral, cognitive, and social) present themselves in the sustainable clothing industry. The manuscript definitely provides new information which is significant but requires some improvement.

  • Thank you for this comment as it encouraged us to refine the paper more. We are pleased to hear that you see contribution in our findings. We understand that the manuscript necessitates more work and therefore we have carefully taken all feedback and improved flaws pointed by you and the other reviewers. We hope that this revised version reads better, and we have corrected identified shortcomings.

My comments and suggestions are as follows:

Abstract

Abstract is well written and captures the main highlights in a very lucid manner. Only suggestion for authors is to arrange the keywords alphabetically.

  • Thank you. We have now rearranged the keywords alphabetically.

  1. Introduction

Authors need to strengthen the introduction section further. I suggest authors introduce the customer perspective on recycling and reuse in general without a definition which could be provided.

  • We fully agree: therefore, we have added a very brief explanation on this. We have now added the following: “-- customer-centric perspectives that focuses on examining how customers consume and perceive circular, recycled or reused offerings.”

On page 2 line 48 authors mention that “….customer experience has been identified as a compelling antecedent in various business contexts….” but they do not tell their readers what customer experience means and how is relevant in this specific context. Authors may want to read and cite following recently published paper that covers customer experience in detail:  https://doi.org/10.1016/j.jretconser.2021.102798

  • Thank you for this important comment and the recommendation build on  the interesting article. We have now added a brief, yet explicit definition and explanation for the customer experience concept, as follows: “-- conceptualized as customers’ subjective, contextual perceptions on offering [11,12] (Kranzbühler et al., 2018; Anshu et al., 2021) 
  • We have also emphasized the importance of customer experience in CE context, in the following way: “Likewise, scholars have emphasized customer experience in achieving satisfied and loyal customers [12](Gentile et al., 2007) who are willing to pay [13](Sachdeva and Goel, 2015), thus making it pivotal also for CE businesses.” 

 In the later section of the introduction authors have identified and mentioned the key gaps but this must be followed by clearly stating the novelty of the present study. Study's objectives can also be clearly stated through research questions in the introduction section.

  • We fully agree with you: the novelty of the study and contribution needs to be clearly spelled out. Therefore, we have aimed to improve this section in our introduction accordingly: there is a new paragraph that concludes and explains the gaps (the references are indicated earlier and before), and contributions are given in more sharp manner.
  • “To summarize the extant gaps in the relevant literature, the CE and environmental sustainability research has neglected customer perspective and particularly customer experience that would explain how circularity and circular products are experienced by the customers themselves, whereas customer experience research has implicitly focused to analyzing linear products without particular focus on circular products, and is therefore lacking specific understanding of how customer experiences are constructed and manifested in circular products.

Due to the abovementioned research gaps and the need for new knowledge, we pose the following research question: “How are CE products (recycled vs. reused) experienced by consumers in the clothing industry per experiential dimensions?” To answer this question, we conduct a qualitative explorative interview study among consumers, map different dimensions of their experiences with recycled and reused clothes to examine how dimensions manifest, and then through comparison to synthetize a more general model that displays how circular products are experienced by consumers. 

This paper makes the following contribution to literature. First, it aims primarily contribute to sustainability and CE research, by building a micro-consumer-level understanding of CE and introduction the experience approach and the consequent subjective individuals’ perspectives on CE business: it develops a conceptual map on how consumers experience circular offerings according to the five experiential dimensions. The contribution here is the model that displays how customers experience circular products through sensing, feeling emotions, cognitive processing, behaving and interacting socially.

  1. Theoretical Background

This section is well written but I would suggest that on page 5 after line 207 authors may want capture the ethical judgement of customer to highlight recycle and reuse. Term ethical judgement along with religiosity for the purpose of environmental sustainability has been captured in the following paper: https://doi.org/10.1016/j.techfore.2021.121094

  • Thank you – this was a good insight. Based on your comment, we realized that ethical judgement aspects are emergently identifiable in our study (in the intersection of cognitive, social, and emotional dimensions) and we have added this aspect to the paper, therefore. Thank you pointing this out. We did not however, put this to theoretical part, but to Discussion part, where we discuss the interaction between experiential dimensions: we interpreted that ethical judgement seem to connect to the interaction of several experiential dimensions: consumers feel and know “what would be right clothes to be consumed” and behave accordingly, when they reflect their choice of circulated clothes. We hope that this insight triggered by your comment works well.

Authors may want to give a careful read to this section and entire manuscript for identifying typos that may be there.

  • It is unfortunate to hear that there were some typos after the language editing and proofreading service we have used. We have now read the manuscript carefully through, and corrected all typos we found.

Methods

Methods and data analysis is the real strength of this paper. However, I have some issues that authors may want to explain and incorporate in the revised version of their manuscript.

On page 6 line 246 authors have mentioned that empirical data comprised 16 consumer-customer interviews of two case companies. Since the sample size of 16 seems too small therefore, I suggest authors provide some justification for the understanding of readers on how this is appropriate for this type of research.

  • We understand your concern but would like to highlight that our applied qualitative research design where for example saturation guided us to assess that we have enough data. In the revised version, we explain this aspect in the following way:

“As our research design was qualitative and explorative, the data set of 16 intensive, yet content-rich interviews were considered an appropriate amount of data for examining experience dimensions and how they manifest. In qualitative studies on customer experiences, the number of interviews have been even lesser that 10 interviews (see e.g.  Dube & Helkkula, 2015). Here the data collection and amount of data was guided also by saturation, which refers to continued sampling until no new substantive information is collected [48](Miles and Huberman, 1994). The saturation point was reached at the sixth interviewee from the reused clothes group and seventh interviewee from the recycled clothes group. However, at least one more interview was conducted to ensure data saturation was achieved. For example, when inquiring the interviewees’ sensory experience on recycled clothes, most of the factors regarding style, aesthetics, and comfort were mentioned already in the first few interviews.”

Also on same page line 265 and 266 authors mention that “…..the interviewees’ ages ranged between 20 and 43 years, and the gender count was divided somewhat equally between males and females…” Authors may want to provide some explanation on why the age groups and gender?

  • Thank you for this remark: from your comment we realized that we have aimed to balance the gender but did not aimed to control the age of interviewees, nor choose them based on their age. In the revised manuscript we now explain this and give also more background information per each interviewee in the newly added Table 1 (that displays gender, age and nationality):

“The interviewees’ ages ranged between 20 and 43 years, and the gender count was divided somewhat equally between males and females. While the gender distribution was intentional to cover the input from both genders, the interviewees’ age was not controlled. The detailed background information is provided in Table 1.”

  1. Results

This section is the real strength of this paper but I suggest authors highlight the value of the paper in few lines at the end of this section. Authors may want to elaborate on this by explaining …how and why?

  • We are happy to hear that our results are convincing. We agree that there was a need to clarify the value of our results. We however felt that the optimal place for this clarification could be the beginning of the discussion section as there as is also a summary/synthesis where we highlight some of the most important results. We now argue that our results enhance the understanding of customers’ experience of circulated products of the clothing industry and enhance the understanding of customers’ motivation to engage in recycled and reused products and adoption of CE products. We hope that this insert works well.
  1. Discussion and conclusion

These sections are nicely written and explained. I don’t see any problem with this. Wish authors all the best.

  • Thank you again. We have done some minor revisions to discussion and conclusions well and hope you find these improvements valuable as well.

Reviewer 3 Report

The article untitled “Customer experience in circular economy: Experiential dimensions among consumers of reused and recycled clothes” focuses on an interesting topic with potential to contribute to previous research. The article is well written and poses some interesting questions. However, after reading the article, results fall below expectations. So I would like to mention some shortcomings that the authors should address in order to publish their manuscript:

  1. The contribution to CX literature is limited, since the article only identifies some CX responses that are context specific inside the theoretical framework, already designed by previous research that differentiates between sensorial, affective, cognitive, behavioural and social responses. The authors state that some responses are interrelated but they do not state how exactly this happens or what this add to previous research.
  2. The definition of social experience is too restricted. The authors focus on social experiences as those responses related to self-esteem. However, these are not the only social responses that consumers may have. Social responses refer to how consumers relate to others and their social environment (Gentile et al., 2007; Schmitt, 1999; Becker and Jaakkola, 2020). So, some examples, not related to self-esteem, could be feelings of shairng with a friend, feel connected to a green reference group, solidarity, etc.
  3. My main concerns are methodological. The study is based on qualitative research with in-depth semistructured interviews. The authors do not explain the duration of these interviews, how they were developed, who conducted them, do not provide information about the sample, and do not attach the script used. These aspects are easy to solve. However, from my point of view 16 interviews to consumers are too few to give rigor and reliability to the findings addressed. Some findings were only supported by one interviewee for example. The authors state that they reach saturation with this sample, but at the view of results, it does not seem so to me.
  4. Results just focus on establishing CX, without referring to its causes, such as touchpoints, stimuli or moderators. So, the study falls short into the readers’ expectations. This is not applying the customer experience paradigm to the consumption or use of used and recycled products, but just identifying experiential responses.  
  5. In some parts of the result section, it seems that the authors propose some rationale for their findings, such as the fact that an Indian interviewee has unlearned his prejudices because of a cultural change. However, stating this may seem adventurous. It is doubtful that the authors can establish an asseveration like this properly because very few interviewees support their finding.

To sum up, my main concern is the reduced sample, which makes me advice against publication. Still, I hope that my comments are useful for the authors and wish them all the best with their research project.

References:

Becker, L., & Jaakkola, E. (2020). Customer experience: fundamental premises and implications for research. Journal of the Academy of Marketing Science48(4), 630-648.

Gentile, C., Spiller, N., & Noci, G. (2007). How to sustain the customer experience: An overview of experience components that co-create value with the customer. European Management Journal25(5), 395-410.

Schmitt, B. (1999). Experiential marketing. Journal of Marketing Management15(1-3), 53-67.

Author Response

The article untitled “Customer experience in circular economy: Experiential dimensions among consumers of reused and recycled clothes” focuses on an interesting topic with potential to contribute to previous research. The article is well written and poses some interesting questions.

  • Thank you for this encouraging comment: we are pleased to hear that you see contribution potential in our paper and consider our paper well written.

However, after reading the article, results fall below expectations. So I would like to mention some shortcomings that the authors should address in order to publish their manuscript.

  • We agree that the article manuscript can be improved, and we are thanking for the chance to develop our paper further. We have now put our effort in providing the version where we have addressed to the identified shortcomings, pointed by you and the other reviewers, and we hope that this version articulates contribution more exactly and reads even better.

The contribution to CX literature is limited, since the article only identifies some CX responses that are context specific inside the theoretical framework, already designed by previous research that differentiates between sensorial, affective, cognitive, behavioural and social responses. The authors state that some responses are interrelated but they do not state how exactly this happens or what this add to previous research.

  • Thank you for this criticism. We would like to highlight that our primary contribution is addressed to environmental sustainability and circular economy research as our study aims to strengthen the customer perspective in CE research by applying CX lenses to this research stream. Only secondarily we aim to contribute to CX literature but believe that our study does generate contribution to this literature as well, as our analysis develops understanding of CX on circulated products of two types (recycle and reuse business models) whereas the conventional CX research mainly (implicitly) concerns linear products. However, we agree that the stronger contribution goes to CE and sustainability research, that is the main focus of the target journal, Sustainability.
  • Based on your comments we have now sharpened how we articulate our intended and realized contributions, to both CE and sustainability literature and CX literature.
  • The primary focus of our investigation was on how each experience dimension manifests in circular products, and the interaction between the dimension came visible in more emerging way. Therefore, we fully agree that we cannot say how exactly the interaction happens. We believe that our findings unmasking interconnections (and we explain some of them) can encourage further research to study in more detail how interaction between dimensions happen. We are now more clearly stating that interrelation between experiential dimensions deserves more research, and argue this in future research section.
  • To argue our contributions clearly, argumentation in the introduction section is revised accordingly, as follows:

“To summarize the extant gaps in the relevant literature, the CE and environmental sustainability research has neglected customer perspective and particularly customer experience that would explain how circularity and circular products are experienced by the customers themselves, whereas customer experience research has implicitly focused to analyzing linear products without particular focus on circular products, and is therefore lacking specific understanding of how customer experiences are constructed and manifested in circular products.

Due to the abovementioned research gaps and the need for new knowledge, we pose the following research question: “How are CE products (recycled vs. reused) experienced by consumers in the clothing industry via the experiential dimensions?” To answer this question, we conduct a qualitative explorative interview study among consumers, map different dimensions of their experiences with recycled and reused clothes to examine how each five dimensions manifest, and then through comparison to synthetize a more general model that displays how circular products are experienced by consumers, per each five dimension.

This paper makes the following contribution to literature. First, it aims primarily contribute to sustainability and CE research, by building a micro-consumer-level understanding of CE and introduction the experience approach and the consequent subjective individuals’ perspectives on CE business: it develops a conceptual map on how consumers experience circular offerings according to the five experiential dimensions. The contribution here is the model that displays how customers experience circular products through sensing, feeling emotions, cognitive processing, behaving and interacting socially.  Second, it develops detailed empirical knowledge of customer experience in two circular business models—recycle & reuse and redistribute— to explain how recycled and reused products differ from a customer’s perspective.”

  • The conclusion section is also revised as follows, to explain our contributions:

”Furthermore, our study contributes secondarily to customer experience and experience marketing pioneered by Schmitt [26] (1999) and Brakus et al. [25](2009), as well as green consumerism research, as we have used the established experience dimension approach in novel, sustainable and circular product settings. Our results explain how circular products are experienced per dimension, and the proposed empirical-based dimensional model displays what constitutes CE customer experiences. These insights build a new understanding on how customers experience circular products that is missing from the extant linear-product oriented customer experience literature.” 

The definition of social experience is too restricted. The authors focus on social experiences as those responses related to self-esteem. However, these are not the only social responses that consumers may have. Social responses refer to how consumers relate to others and their social environment (Gentile et al., 2007; Schmitt, 1999; Becker and Jaakkola, 2020). So, some examples, not related to self-esteem, could be feelings of shairng with a friend, feel connected to a green reference group, solidarity, etc.

  • We see your point and we have now rephrased how we define social experience dimension. Our results concern mostly on social interactions, emerging from (consuming) circulated clothes: therefore it was valuable that you pushed us to sharpen and clarify how the social experience dimension is explained in theory section. We agree that the earlier definition was too narrow, thank you for pointing this out, and we believe that with the more sharp definition of social experience in the front part, also our findings on social dimension are more understandable.

My main concerns are methodological. The study is based on qualitative research with in-depth semistructured interviews. The authors do not explain the duration of these interviews, how they were developed, who conducted them, do not provide information about the sample, and do not attach the script used. These aspects are easy to solve. However, from my point of view 16 interviews to consumers are too few to give rigor and reliability to the findings addressed. Some findings were only supported by one interviewee for example. The authors state that they reach saturation with this sample, but at the view of results, it does not seem so to me.

  • We understand your concern on the amount of data, but here we would like to highlight here that our approach was explorative and qualitative. The earlier published studies on customer experience have been grounded even lesser amount of data (e.g. 7 informants in paper by Dube & Helkkula, 2015). Our research goal was to explore and examine how the five experiential dimensions manifest when consumers experience circulated products (clothes). During the research process we faced saturation, meaning that no new, particularly relevant information emerged from interviews, after the seventh interview per category, regarding the dimensions. Therefore, conducting more interviews would not have significantly changed the main experiential triggers identified.
  • In the revised version, we explain this aspect in the following way:

“As our research design was qualitative and explorative, the data set of 16 intensive, yet content-rich interviews were considered an appropriate amount of data for examining experience dimensions and how they manifest. In qualitative studies on customer experiences, the amount of interviews have been even lesser that 10 interviews (see e.g.  Dube & Helkkula, 2015). Here the data collection and amount of data was guided also by saturation, which refers to continued sampling until no new substantive information is collected [48](Miles and Huberman, 1994). The saturation point was reached at the sixth interviewee from the reused clothes group and seventh interviewee from the recycled clothes group. However, at least one more interview was conducted to ensure data saturation was achieved. For example, when inquiring the interviewees’ sensory experience on recycled clothes, most of the factors regarding style, aesthetics, and comfort were mentioned already in the first few interviews.”

  • We agree that much more background information on interviewees is needed. Please find the newly added Table 1 that displays ages and genders of interviewees, as well nationality.
  • We have also now explained the duration and who has conducted the interviews. We give also more information on our strategy to capture “fresh” experience at site and via a structured guide that covers all experiential dimensions.
  • We have now added also much more clear and explicit explanation on how saturation was reached (please see e.g. 3.1. and 3.2.).
  • We also would like to express that in the paper we have used “one” interviewee often as an example. So, there is often much more support for the finding than one single interview. We have therefore rephrased many sentences, to avoid misreading that there would be only one respondent, in cases where “one” has been used as an example, instead.

Results just focus on establishing CX, without referring to its causes, such as touchpoints, stimuli or moderators. So, the study falls short into the readers’ expectations. This is not applying the customer experience paradigm to the consumption or use of used and recycled products, but just identifying experiential responses.  

  • We have now clarified our research purpose and goals: We are not aiming to primarily contribute to customer experience paradigm (as we have not targeted our study to CX focused journal but sustainability and CE journal). We believe that sustainability researchers benefit from learning customer perspective and CX approach that we are applying in this study. However, we believe that linear-product oriented CX research paradigm can learn some insights from our empirical study on experiencing circular products. We hope that we have phrased this more clearly in the revised version and argue that we primarily contribute to sustainability and CE literature by applying CX dimension lenses, and secondarily to CX literature by exploring how dimensions are manifested when consumers experience circulated products.
  • We fully agree with you that it is relevant to study also touchpoints, stimuli and moderators, also for circular offerings. Thank you for bringing this into our consideration, this was a good point. We have therefore now put the aspects you have raised (the need to study touchpoints, stimuli and moderators in more detail) into future research section. We hope this encourages researchers to study further how experiences of circular products are triggered and what moderates experiences.

In some parts of the result section, it seems that the authors propose some rationale for their findings, such as the fact that an Indian interviewee has unlearned his prejudices because of a cultural change. However, stating this may seem adventurous. It is doubtful that the authors can establish an asseveration like this properly because very few interviewees support their finding.

  • We also understand your concerns on making interpretations from one, single interview. We would like to highlight that our intention, due to qualitative research design, was to explore the dimensions and not statistically validate them. But we fully understand that we need to carefully phrase the emerging findings from our qualitative data. Therefore, we have phrased the paragraph in the following way, more conceptually, the focus being on the potential pattern and rationale for why Indian and European-originated consumers’ social interaction around circulated clothes has been so different and how this reflects to their experiences with circular products.

To sum up, my main concern is the reduced sample, which makes me advice against publication. Still, I hope that my comments are useful for the authors and wish them all the best with their research project.

  • Thank you for all your comments, we have aimed to respond carefully to your and other reviewers’ comments and suggestions for improvement, and we hope that the revised version is more convincing and clearer.